# “It Goes Hand in Hand with Us Trying to Get More Kids to Play” Stakeholder Experiences in a Sport and Active Recreation Voucher Program

**DOI:** 10.3390/ijerph20054081

**Published:** 2023-02-24

**Authors:** Bridget C. Foley, Natalie Turner, Katherine B. Owen, David Cushway, Jacqueline Nguyen, Lindsey J. Reece

**Affiliations:** 1SPRINTER, Prevention Research Collaboration, Charles Perkins Centre, Sydney School of Public Health, Faculty of Medicine and Health, The University of Sydney, Sydney, NSW 2006, Australia; 2Office of Sport, New South Wales Government, Sydney Olympic Park, NSW 2127, Australia

**Keywords:** financial incentive, fiscal, cost, organizational capacity, active kids, sports management, government intervention, large-scale program

## Abstract

Vouchers that reduce the cost of sport and active recreation participation have been shown to increase children’s and adolescent’s physical activity levels. Yet, the influence of government-led voucher programs on the capacity of sport and active recreation organisations is unclear. This qualitative study explored the experiences of stakeholders in the sport and recreation sector that were engaged in implementing the New South Wales (NSW) Government’s Active Kids voucher program in Australia. Semi-structured interviews were conducted with 29 sport and active recreation providers. Interview transcriptions were analysed by a multidisciplinary team using the Framework method. Overall, participants reported that the Active Kids voucher program was an acceptable intervention to address the cost barrier to participation for children and adolescents. Three main steps influenced the capacity of organisations to deliver their sport and recreation programs and the voucher program: (1) Implementation priming—alignment of the intervention aims with stakeholder priorities and early information sharing, (2) Administrative ease—enhanced technology use and establishment of simple procedures, and (3) Innovation impacts—enablement of staff and volunteers to address barriers to participation for their participants. Future voucher programs should include strategies to enhance the capacity of sport and active recreation organisations to meet program guidelines and increase innovation.

## 1. Introduction

### 1.1. Sport and Recreation for Health

The World Health Organization has called for stronger partnerships between stakeholders to enable more children and adolescents to be physically active more often [1]. The sport and recreation setting is well recognised as important for promoting physical activity participation and health [2,3]. Participation in structured physical activities, including sports and active recreation, provides additional physical health, social, and psychological benefits, compared to the benefits of physical activity in other settings. These benefits include reducing symptoms of depression; less internalising of problems; improved self-esteem and confidence; improved self-discipline; and enhanced time management, teamwork, and leadership skills [4,5,6]. Structured physical activity programs can provide children with opportunities to participate in different types and higher intensities of physical activity, which is associated with additional benefits [7]. Critically, participation in structured physical activities during childhood and adolescence is associated with greater physical activity participation later in life and the prevention of non-communicable diseases and mental health conditions [8,9,10]. However, participation is unequal, with children and adolescents from low socioeconomic and minority groups less involved and therefore missing out on the many benefits [11].

### 1.2. Australian Sport and Recreation Context

Organisations in the sport and recreation sector are vital stakeholders in the promotion of physical activity, particularly structured programs. Structured physical activity programs delivered by organisations in the sport and recreation sector include team sports (e.g., football, netball, basketball, and hockey); individual sports (e.g., swimming and athletics); and structured recreation (e.g., dance, martial arts, and bush skills). The typical operating model in the Australian context has three tiers of sport and recreation organisations: National Sporting Organisations (NSO) and State Sporting Organisations (SSO) govern and support affiliated community-based, grass roots clubs and associations to deliver structured physical activity programs. Other operating models in the sport and recreation sector include businesses or independent not-for-profit organisations that an NSO or SSO does not govern. The objective of most sport and recreation organisations is to provide enjoyable, accessible, inclusive, and affordable structured physical activity programs while maintaining financial sustainability [12]. Typically, these organisations are not-for-profit, gain revenue from memberships, and rely on a large volunteer workforce to achieve their objectives [12].

Organisations in the sports and recreation sector facilitate the participation of over 3.5 million Australian children in structured physical activity outside of school time each week [13]. Australian children participate in 4–6 h of structured physical activity each week [14]. Organisations delivering structured physical activity programs need to cover operational costs, including staff wages, public liability insurance, goods and services taxes, and facility hire costs, which influence the capacity and subsequently drive-up costs for participants. Research shows that Australian families spend approximately AUD 1250 per child per year to enable children to participate in structured physical activity outside school time [14]. The high cost associated with structured physical activity programs is a barrier to participation and unfairly impacts girls and socioeconomically disadvantaged children [14,15,16]. Reducing or removing the cost barrier to structured physical activity participation is critical to increasing physical activity participation [15,16,17]. There is a need to build capacity in sport and recreation organisations to provide affordable opportunities for all children and adolescents to participate in structured physical activity programs [1,18].

### 1.3. Capacity of Sport and Recreation Organisations

Capacity is broadly defined as an organisation’s ability to achieve its objective with available assets and resources [19]. The ability of stakeholders in the sector to increase participation in their structured programs can be influenced by multiple contextual factors such as geographic location; organisational size; participant demographics (e.g., age and gender); and the activity type (team sport, individual sport, and structured recreation). Acknowledging contextual differences, Doherty, Misener, and Cuskelly developed a multidimensional framework to understand capacity in community sports clubs [19]. The framework includes five dimensions in sports clubs that influence the organisational capacity: human resources (staff, volunteers, and members); finance (memberships, fundraising, government support, and other revenue); infrastructure (information technology and facilities); planning and development (strategies, tailored initiatives, marketing, and promotion); and external relationships (partnerships and relationship with the government) [19]. Changes to each dimension can profoundly affect the already stretched capacity of organisations to provide enjoyable, accessible, inclusive, and affordable structured physical activity programs for all. For example, if the number of volunteers in an organisation declines, they may need to reduce participation opportunities or increase fees to pay staff to fill the volunteer’s role. This vulnerability to change contributes to inadequate innovation in the sport and recreation sector to increase physical activity participation.

### 1.4. Financial Incentives

Governments can use economic tools (such as taxes and subsidies) to influence the costs associated with structured physical activity participation [20,21,22]. Financial incentives such as subsidies and vouchers provided to individuals to motivate participation in structured physical activity are becoming increasingly popular; however, evidence on the effectiveness and sustainability of these interventions is mixed [20,22,23,24,25]. Most financial incentives have been studied among adult populations, finding that even short-term financial incentives can lead to long-term increases in physical activity [20]. The few studies into the role of financial incentives among children and adolescents have been conducted and have demonstrated mixed effects. Internationally, Canada and the United Stated (specifically, Los Angeles) have trialled refundable tax credits as incentives to promote physical activity, which did not increase children’s physical activity levels [26,27,28,29]. Conversely, voucher programs trialled in Wales and Australia have demonstrated positive effects on physical activity in small- and large-scale studies, respectively [14,30,31]. Other financial incentive programs in Germany have demonstrated no effect on children’s involvement in sports [25,32]. In this emerging field of research, studies of financial incentive studies have focused on the reach and effect of financial incentive programs among children and adolescents. Limited details regarding the implementation processes and stakeholder involvement in financial incentive interventions for children and adolescents have been reported. The effect of financial incentives on the capacity of organisations within the sport and recreation sector is often overlooked [3,33]. Doherty et al. identified finances as a key dimension of organisational capacity, and financial incentives are likely to impact on the capacity of structured physical activity providers to offer inclusive and affordable programs. There is a need to understand the intended and unintended consequences of financial incentive interventions on the organisational capacity of structured physical activity providers. Understanding how the government and stakeholders in the sport and recreation sector implemented this large-scale voucher program will have implications for the design and delivery of future interventions.

## 2. Materials and Methods

### 2.1. Study Design and Aim

This cross-sectional qualitative study aimed to understand the implementation process of an Australian financial incentive (Active Kids voucher) program from the perspective of sport and recreation organisations and explore the impacts of the voucher program on their organisational capacity.

### 2.2. Setting of the Study

In 2018, the New South Wales (NSW) government in Australia launched a four-year voucher program titled ‘Active Kids’ that aimed to reduce the cost of participation in structured physical activity programs outside of school time for children and adolescents [34]. All school-enrolled children (4.5–18 years old) that resided in NSW (2.1 million) were eligible for one Active Kids voucher per year valued at AUD 100 (USD 70). Parents and caregivers applied for Active Kids vouchers online and could redeem them with approved organisations delivering structured programs which lasted at least 8 weeks and included moderate-to-vigorous physical activity. The voucher could not be used for sports equipment or clothing. The structured physical activity providers that redeemed the vouchers were key stakeholders in the implementation of this program and did not receive reimbursement for redeeming a child’s Active Kids voucher.

To register as a provider with the Active Kids program, organisations must deliver a sport or structured active recreation program and adhere to the Active Kids Provider Guidelines [35]. Registered Active Kids providers redeem Active Kids vouchers from participants by entering the child’s voucher number, name, and date of birth into a bespoke centralised government platform. The AUD 100 value of each voucher redeemed is then deposited into the provider’s bank account from the NSW government. Over 550,000 children’s Active Kids vouchers were redeemed with a registered Active Kids provider in 2018 [14]. The voucher was not designed to provide additional revenue to Active Kids providers; their registration fees should be consistent whether or not children redeem a voucher.

Evaluation of the Active Kids program has shown that children who redeemed an Active Kids voucher to participate in a structured physical activity program increased their days achieving physical activity guidelines from 4 days per week at registration to 5 days per week 6 months after using a voucher [14]. The most common types of activities school-aged children redeemed their voucher for were soccer (football), netball, swimming, multi-sport, dance, rugby league, gymnastics, basketball, Australian rules, and rugby union [14]. The Active Kids voucher reduced the cost of registration or membership fees, supporting, on average, 20% of the annual structured physical activity participation costs [14]. Whilst the experiences of children that use an Active Kids voucher have been comprehensively examined, the experiences of the stakeholders (Active Kids providers) involved in the implementation are a critical process evaluation component that are yet to be explored [36]. This large-scale, government-led voucher program has the potential to impact the capacity of organisations to provide structured physical activity programs for children and adolescents.

### 2.3. Participant Sampling and Recruitment

In June 2019, the NSW Government Office of Sport database of Active Kids providers contained 10,037 approved Active Kids providers offering activities to children across the state. The Office of Sport staff selected 81 organisations from the Active Kids database for inclusion in this study using a quota sampling technique, randomly selecting providers from their database until each quota was reached. The sample of providers was selected based on the number of voucher redemptions recorded in the database (small < 50, Medium 50–100, and Large > 100 vouchers); location where they delivered most of their activities (metropolitan/regional); and SSO affiliation (Yes/No/SSO). SSOs were defined by whether the provider organisation was a recognised SSO in NSW, and affiliation was defined as a grass roots club/association affiliated with a recognised SSO [37]. Further details on the selection process were not recorded by the Office of Sport staff. The participant recruitment flow is shown in Figure 1.

The selected Active Kids providers were invited to participate in the study by email from the Office of Sport staff, which included the participant information sheet and contact details of the research team. After one reminder email, those who did not respond to the email were followed up with a phone call using the phone numbers they included in the registration form. More than half (n = 44) of the invited provider’s phones were disconnected or did not include a phone number in their registration. Participants who replied to the invitation email and provided written consent for this study were scheduled for interviews with the researchers. Verbal consent was also obtained from all participants before their interviews commenced. Participant details and the length of their interview are presented in Table 1. Broad categories for sport type, participant’s role, organisation size, and location have been used to maintain the anonymity of the participants (Table 1).

### 2.4. Semi-Structured Interviews

The topic guide was codeveloped in partnership with policymakers at the NSW Government Office of Sport to elicit provider’s experiences delivering the Active Kids program during a 30-min telephone interview. The semi-structured interview guide was developed using the Doherty, Misener, and Cuskelly (2013) multidimensional framework of capacity in grass roots sports clubs [19]. The topic guide asked stakeholders about the impacts of the Active Kids program on each dimension of the framework, namely human resources, finances, infrastructure, planning and development, and external relationships [19]. The topic guide also asked stakeholders about their reasons for becoming an Active Kids provider, their understanding of the program, and their opinions on what worked and what did not in the implementation process (Appendix A). This study contributes to the process evaluation of the Active Kids program as part of the larger program evaluation detailed elsewhere [36].

### 2.5. Data Collection

BCF led the data collection by conducting 25 telephone interviews and supervised NT conducting 4 telephone interviews. All interviews followed the semi-structured topic guide (Appendix A). The interviews were voice-recorded and transcribed verbatim by an Australian transcription company. Participants could review the transcription before analysis; four participants opted to be sent their interview transcript with zero revisions provided. The names of the individual and their organisation were removed before the analysis. Active Kids providers are not identifiable in this study.

### 2.6. Data Analysis

A multidisciplinary research team used the Framework Method for data analysis to ensure trustworthiness [38,39]. The Framework Method involves six steps after transcription. (1) Familiarisation: BCF and NT emersed themselves in the data by thoroughly reading and checking the transcripts against the audio-recorded interviews. BCF was heavily engaged in the evaluation of the Active Kids program, and NT had no prior exposure to the program before the analysis, which reduced researcher bias. (2) Coding: BCF and NT independently read all transcripts, recording their impressions through open coding. Their open coding involved underlining key segments of the text and annotating the margins with any preliminary impressions. (3) Developing an analytical framework: Building from the Doherty, Misener, and Cuskelly (2013) multidimensional framework of capacity in grass roots sports clubs [19], the interview questions, and the annotated notes, both researchers discussed the key concepts that had emerged from the data. BCF and NT then met with LR and KO to discuss the data’s key concepts and recurrent themes. Using a whiteboard and coloured pens, the researchers devised a set of dependable codes and subcodes, each with definitions, forming the initial analytical framework for the study. (4) Applying the framework: Systematic application of the major codes was done independently by BCF and NT using NVivo software (NVivo, RRID:SCR_014802) on three transcripts. By comparing their application of the analytical framework, the definitions of each code were further defined and updated by grouping and creating codes. Revising, applying, and redefining the framework was iterative until a final analytical framework was confirmed (Figure 2). The final coding framework included the five dimensions of sports organisation’s capacity and key concepts that emerged from the data regarding the implementation of the Active Kids program [19].

Once the interrater reliability reached 80%, BCF and NT independently coded all transcripts in NVivo using the brief definitions of the codes to uphold consistency. Illustrative quotes were marked using the “annotation” feature. (5) Charting the data: Once coding was completed, data were interpreted in a Framework matrix with codes horizontal and cases vertically. This matrix allowed patterns, differences, and similarities to be identified and explored by the researchers. The charted data enabled understanding of the relationships between the implementation of the Active Kids program and capacity dimensions, extending understanding beyond the coded data. (6) Interpreting the data: Ongoing consultation between the multidisciplinary team occurred during the interpretation of the data to ensure trustworthiness and reduce the bias of the analysis. After the analysis was completed, the findings were presented to authors working on the Active Kids program implementation (DC and JN), which confirmed the findings reflected anecdotal feedback from providers as part of their daily practice. Three predominant themes were identified that influenced dimensions of organisational capacity during the implementation of the Active Kids program.

## 3. Results

Twenty-nine Active Kids providers participated in the semi-structured interviews, which had an average duration of 31 min (Table 1). Participants represented 15 large organisations that had redeemed over 100 Active Kids vouchers, 7 medium organisations that had redeemed 50–100 vouchers, and 7 small organisations that had redeemed <50 vouchers. About half (56%) of the participants were operating in metropolitan areas, and 76% were affiliated with an NSO/SSO (Table 1). The structured physical activities delivered by participants included team sports (n = 13), individual sports (n = 7), structured recreation (n = 7), and disability sports (n = 2) (Table 1).

Implementation of the Active Kids program had varied effects on the five dimensions of the Active Kids provider’s capacity [19]. The structured analysis process identified three themes that were required to support the organisational capacity of structured physical activity providers during the implementation of the Active Kids program: (1) Implementation priming, (2) Administrative ease, and (3) Innovation impacts (Figure 3). The three themes and seven subthemes in Figure 3 are detailed with illustrative quotes, providing evidence of what worked and what did not for various stakeholders involved in the Active Kids program in NSW. The step design in Figure 3 was used to demonstrate the foundational processes required to achieve innovation impacts during implementation. Relationships between stakeholders and the NSW Government Office of Sport at the time of implementation strongly influenced the organisational capacity across the three themes and enabled progress from one step to the next (Figure 3).

### 3.1. Implementation Priming

#### 3.1.1. Alignment between the Program Aim and Sector Activities

The Active Kids providers perceived the voucher program overall as a good investment by the government to reduce the cost of structured physical activity for children. The government program aims strongly aligned with those of the providers to remove barriers to participation. This common goal contributed to providers’ acceptability of the Active Kids program.

“*It goes hand in hand with us trying to get more kids to play, so I think it’s been a winner*.” Committee member, Medium Affiliated organisation

Most providers reported that the program’s requirements suited what they were already delivering for children. Providers reported they typically offer inclusive participation opportunities, where anyone could join their activity, and their voucher activities were no different. Nearly all (n = 28) providers kept offering their usual activities, as they already met the government criteria. The program did not require changing the provider’s organisational planning and development. This alignment further contributed to the acceptability of the Active Kids program among providers.

“*All of our programs were term-based anyway, they run for greater than the minimum requirement of eight weeks for the Active Kids program. So, it wasn’t necessarily a change for us*.” Manager, Non-affiliated, large organisation

One SSO that traditionally offered physical activity programs mainly to older age groups reported using the Active Kids program to encourage their affiliated clubs to engage more kids.

“*In the past, a majority of the members are actually more veteran members, and if the sport wants to grow and become larger, it’s a good way to attract more younger members*.” Manager, SSO

Cost reduction through the Active Kids voucher was perceived as a good support for families. Providers reported that staff and volunteers agreed with the need to reduce participation costs for members and were motivated to encourage voucher use. Providers said the program made it easier for families to enrol children in sports programs and keep them playing sports.

“*The older kids, they get to 14, 15 and they start to drop out and forget about sport. But parents are now encouraging those kids to stay in sport and saying hey, we’ve this $100 voucher, then those kids are going okay, it’s not costing my parents anything*.” Committee member, Affiliated small organisation

#### 3.1.2. Program Awareness, Knowledge, and Understanding

Awareness of the program among providers was achieved in a variety of ways. Providers reported first hearing about the Active Kids program through organisational emails (n = 11), communication from the government (n = 5), communications from other sports providers in the sector (n = 5), media stories/articles (n = 4), or from parents who wanted to redeem vouchers for their activities (n = 4). How providers first heard about the program was influenced by their external relationships. SSOs and affiliated organisations that had existing relationships with the Office of Sport heard about the program before it launched, while non-affiliated organisations found out about the Active Kids program after it launched to the public, resulting in not having the human resources (staff) and infrastructure (information technology (IT) systems) available to redeem vouchers from children.

“*The league actually introduced it to us, saying that it was coming onboard, and we all went through the process of getting signed up and being Active Kids providers… The league did a pretty good job of actually telling us that it was coming.*” Affiliated SO, President

“*I had parents contacting me. Do you have the Active Kids program? I’m like, never heard of it, so I had to look it up. I was definitely encouraged by my clients to do it*.” Non-affiliated, Business owner

All providers reported having a good understanding of what the Active Kids voucher could be used for within their organisation, i.e., membership and registration fees. They reported gaining their understanding of the program from resources on the Office of Sport website, which was either distributed to them or identified by the individual. Few providers (n = 12) were aware of additional resources and developments in the program beyond the voucher itself due to limited communication from the Office of Sport to providers, especially non-affiliated providers (see Table 2). Most providers were aware that the government had announced that a second voucher would be available during 2019 however many heard this through news media rather than stakeholder communications from the government (Table 2). 

“*There was no information that they were offering a second one sent to me. Little things like that. I just don’t think it’s well communicated*.” Business owner, Large, Non-affiliated organisation

### 3.2. Administrative Ease

The NSW Government Office of Sport led the development and implementation of Active Kids, whilst the voucher administration was undertaken through a centralised government platform led by a government department exclusively devoted to Services, Service NSW. This dual responsibility across government departments caused some communication challenges when providers had issues or concerns. Identifying the appropriate government department responsible for resolving an issue was not a straightforward process, which negatively impacted the organisational capacity. The main administrative processes for Active Kids providers were registration in the program and voucher redemption. The Active Kids providers’ relationships with the Office of Sport influenced their organisational capacity to achieve administrative ease.

#### 3.2.1. Registration in the Active Kids Program

Large non-affiliated organisations and SSOs faced unique administration challenges to meet the requests from the government to upload all their affiliated club/association details to the system. Organisations involved in the testing and developing the registration platform for providers did not receive specific remuneration, placing additional resourcing strains on these organisations to meet the government requirements.

“*[the government] hadn’t thought through all the processes to implement it … none of the sports had actually budgeted for the cost to get this implemented because of the staff time to collect information and the API that was required. Even though the government provided it, we as a state sporting organisation still had to pay for the implementation of that through our service providers*.” Manager, SSO

Operationalising the Active Kids program universally across all sports, using existing systems, within the government timeline was not considered straightforward. SSOs reported a sense of obligation from the government to ensure most of their affiliated clubs/associations participated in the program. Following this, non-SSOs felt pressure from others in the sport and recreation sector to be registered so they were not seen as disadvantaging their membership base.

“*We know there were other sports that were signing up for it, so we thought we’d better do it with [our sport]. And it’s a good service, obviously, to the players to be able to get a discount*.” Business owner, Large affiliated organisation

Those who became involved with the program after it had been launched were affiliated with smaller, less-resourced SSOs or independent businesses or franchises. These smaller providers reported the registration process as simple; rather than having an SSO do it for them, those who registered themselves reported the process as being easy to undertake.

“*[The registration process was] no trouble at all. It wasn’t hard to do. There was a whole list of things we had to send through. And then we waited a month and then we got a notification that we were a registered provider*.” Business owner, non-affiliated large organisation

In the program’s second year, providers reported registration and re-registration were simple compared to the initial set-up.

#### 3.2.2. Redeeming an Active Kids Voucher

The voucher redemption process, where providers log on and redeem the voucher through an online government portal, generally exceeded expectations. Active Kids providers reported that the process was simple if they had automated the system or were redeeming vouchers in small numbers.

“*Initially, there was those couple of, let’s call them logistical hurdles. But once they were overcome, it ran very smoothly. It was quick to redeem. It was quick to get the money back into the bank*.” Affiliated SO, regional

Organisations that received large volumes of Active Kids registrations had typically developed sophisticated systems for redemption in partnership with the Office of Sport within the first year of implementation. Large organisations reported the administration changes came at a significant expense to the organisation; however, they deemed this cost worthwhile to ease the administrative burden on their affiliated clubs/organisations and simplify the voucher redemption process. Non-affiliated organisations were still refining the ongoing voucher redemption process, which had mixed effects on organisational resources. Medium/Large organisations that did not have sophisticated online systems in place or did not have the budget to change their registration IT systems expressed frustrations in processing the vouchers manually. Small organisations did not share the same frustrations as medium/large organisations without sophisticated IT systems due to the reduced frequency of the task.

“*It became such a gigantic job that I’ve had to get one of my staff members now to take over [redeeming the vouchers] … its’ an admin nightmare. I now have to pay a staff member to do all the inputting, because I just don’t have time to do it. It’s become an additional cost for the business*.” Business owner, non-affiliated organisation

While, overall, the voucher redemption had been refined by the time the interviews took place, confusion remained among providers when something went wrong during the redemption process. The providers and their affiliated organisations, or individual providers, did not know where to report and solve administrative issues. The different roles of the two government agencies, the Office of Sport and Service NSW, led to confusion in the communication pathways.

“*The level of information that we were provided was pretty much sub-standard. It was difficult because we were told one situation but then our members were told something else. And it just sort of went around in a, in a circle*.” Manager, SSO

### 3.3. Innovation Impacts

Stakeholders reported that implementation of the Active Kids program positively impacted their finances through participation and memberships, improved the motivation and organisational climate (human resources) and strengthened external relationships. There was limited innovation in the planning and delivery of structured physical activity programs by stakeholder organisations, which appeared to be related to the alignment between the program aims and sector activities.

#### 3.3.1. Finances through Participation and Membership

Provider’s perception of the Active Kids program on their membership was largely positive, with no anecdotal reports of reductions from participating Active Kids providers. Some participants had easily accessible records to monitor voucher use (n = 4), while others did not have access to the information or were not monitoring (n = 25) voucher redemption rates in their sport. Anecdotally, eight providers reported increased participation numbers.

“*It’s significant. We’ve seen 25% growth in our club, literally this year. Last year, we took a big step up*.” Affiliated SO, President

Three providers described that the voucher was attracting more family members of their usual participants to begin memberships after the organisation became a provider.

“*Siblings join them who wouldn’t have before. We’ve got one family whose kids are foster care kids, and so they might not have had that opportunity to [play sport together], but now they do because [the Active Kids voucher] gives them that extra bit of financial help to be able to do that*.” Committee member, SSO

Two providers reported they thought the voucher might increase their older children’s participation and retention. However, the majority did not perceive the voucher to impact the number of registrations or memberships at their organisation.

“*I don’t think it’s made an impact in terms of numbers or, even the demographics. I think it’s really just been helping out in our existing members financially*.” Manager, Large organisation

#### 3.3.2. Improved Motivation and Organisational Climate

The Active Kids vouchers stimulated new motivation for some providers and their staff members to reach and engage new participants. Those providers working in socioeconomically disadvantaged areas and/or with low-cost activities most reported this shift in approach. Knowing that families would be assisted financially through the Active Kids program and that membership would not result in less money for essential items empowered staff to encourage families to invest in their child’s sports registration.

“*It’s been a positive impact on staff because, they feel that they can better sell our programs to other people and know the fact that everybody can be involved in these programs, even if they are a little bit socio-economically disadvantaged … it makes staff feel more confident and empowered when talking to people on the phone.*” Manager, Non-affiliated large organisation

“*I think it’s the first time in many years memberships have all been paid up, and I’m not chasing families for money during the year*.” Committee member, SSO

This empowerment of staff and volunteers facilitated further promotional activities to increase participant recruitment. Providers took the Active Kids program as an opportunity to promote their activity, as well as ensure their community members were aware of how to utilise the vouchers. Many providers (n = 16) reported increasing their marketing activity once registered as providers and including Active Kids in this activity promotion.

“*I’ve now added the approved provider logo that the New South Wales government said that we were allowed to add…. certainly, having that logo on our marketing is reducing the barrier to entry*.” Committee member, Medium Affiliated organisation

#### 3.3.3. Strengthened External Relationships

Through delivering the Active Kids program, the collegiality of the sports industry may have adjusted towards increased collaboration between providers within one structured physical activity and/or across different organisations. For example, providers who adopted the program early could assist their peers in registering for the program in the absence of SSO guidance. Providers reported promotional activities were primarily focused on ensuring their community members were aware of the Active Kids program and that engaged children could utilise the vouchers to be active. Providers were still satisfied if their participants had used their voucher with another provider, as long as they had used the voucher. One participant reported holding a forum in her regional community after noticing that local families were unaware of the voucher or how to access it. They conducted sessions to increase the registration of children in any structured physical activity opportunity, not just specifically their sport.

“*In our community hall, we set up three stations of laptops. And we got parents to come in with a group of community people and show them how to use the laptops to get onto their sites to get the access, and then they could print them out there and then, or they could just write their number down and access them later on and lodge it wherever they wanted to*” Committee member, Medium Affiliated organisation

Another organisation reported doing presentations at local schools to increase awareness of their sport and the Active Kids program. Most providers (n = 26) reported asking individuals whether they had an Active Kids voucher when they began the registration process with a new or existing member. Those who actively asked parents and caregivers if they had a voucher would often help people to sign up their children for the Active Kids program so that it could be used when paying their registration fee. Some providers (n = 3) were less proactive in promoting the Active Kids voucher, although they would agree to redeem vouchers when customers presented them. These passive Active Kids providers were the organisations who reported administrative challenges redeeming larger volumes of vouchers.

With the introduction of a second voucher in the latter part of the year, some providers reported developing informal partnerships with other sports for their off season; however, the majority (n = 25) of providers did not report ongoing changes in their partnerships with others in the industry.

“*a lot of the other codes are starting to realise that what we’re training from a skill basis is enhancing their skills and their own codes. Some of the clubs in some of the areas including ours are actually actively promoting to go and play [approved activity] if you want to up your skills. We’re no longer looked at as the enemy anymore. We’re looked at as supplementary…The cross promotion has 100% been influenced by the Active Kids program*” Committee member, Affiliated organisation

Providers that reported leveraging the Active Kids program to achieve these innovation impacts were only able to do so after achieving administrative ease. Often these organisations were part of an SSO or affiliated with an SSO. Challenges with organisational infrastructure and the expenses required to administer the program reduced the capacity of providers to do things differently.

## 4. Discussion

This study was the first to explore the impact of implementing a large-scale financial incentive intervention on the capacity of structured physical activity organisations. As part of a comprehensive evaluation of the Active Kids program, this study provides evidence with practical implications for policy makers planning and designing financial incentive programs in partnership with stakeholders in the sport and recreation sector. The introduction of the Active Kids voucher program initially reduced organisational capacity particularly regarding financial, infrastructure (IT systems), and human resource dimensions of capacity. During ongoing implementation, these challenges were mostly resolved. Three themes were identified from the data that demonstrated practical supports that should be considered when implementing a financial incentive or similar intervention. Firstly, organisations reported intervention priming helped enhance their ability to respond to the government program. Second, support to update administration systems were required, particularly for medium-large organisations and those not affiliated with national or state sporting organisations. Third, stakeholders who were well supported had the capacity to leverage positive impacts from the intervention. Government interventions that require structured physical activity providers to achieve their objectives should include strategies to build their partners’ capacities through financial assistance or training. This study highlights the unintended impacts of a large-scale financial incentive program on implementation partners. It suggests ways to strengthen the capacity of structured physical activity providers for enhanced implementation.

### 4.1. Building Capacity as Part of Government Interventions

Research has shown that sport and recreation sector organisations have limited capacity yet, financial incentive interventions have not previously assessed the impact of the intervention on organisational capacity [12,25,26,40]. Monitoring the impact of interventions on the capacity of structured physical activity providers is important for research translation; and to ensure children and adolescents can participate in physical activity outside of school time [19,41]. In the present study, Doherty et al.’s multidimensional framework for organisational capacity was used to inform data collection and aid the interpretation of the results, providing a deeper understanding of the impacts of the intervention on stakeholders’ capacity [19]. The most critical aspects of capacity for the implementation of the Active Kids program were infrastructure and finances available to achieve administrative ease. Where possible, future programs should allow more time for testing and development of administration or IT systems. Some organisations required substantial support to align their unique administration processes with the government administration requirements. Other studies of organisational capacity among structured physical activity providers have found that human resources were the most critical aspect of capacity for structured physical activity providers [42]. This difference is likely due to the Active Kids program not requiring enthusiastic staff/volunteers to achieve voucher redemption compared to sports participation programs where human resources (e.g., coaches) can substantially influence program outcomes and sustainability. This study adds to previous organisational capacity research and is a first step to guide how governments to avoid potential detrimental effects in the delivery planning and implementation of future financial incentive interventions in the sport and recreation sector.

### 4.2. Engaging a Diverse Group of Stakeholders

The sport and recreation sector comprises a range of different organisations in NSW, including SSOs, non-affiliated organisations, and business owners that provide structured physical activity programs. Traditionally, the sub-national government engage most with SSOs however a broader definition of stakeholders was adopted for the Active Kids program to remove barriers to sport and other types of structured active recreation. This broader definition resulted in a more diverse group of stakeholders being involved in the program, many of whom had not previously had a direct relationship with the Office of Sport. It also increased the number of potential stakeholders that could be engaged as partners in the implementation process. Nichols et al. demonstrated the need for different types of organisations in the sport and recreation sector to receive different types of support to achieve government objectives [43]. This was also observed in the present study with SSO’s and their affiliated organisations reporting the greatest capacity to leverage the Active Kids program. Future sport and recreation interventions should continue to adopt this broader definition to strengthen partnerships with organisations aiming to promote physical activity in the community but consider tailored support for non-traditional stakeholders.

### 4.3. Strategies to Promote Financial Incentives

Details of implementation processes have been under-reported in previous studies of financial incentives encouraging children and adolescents to participate in physical activity outside of school. For the Active Kids program, promotion was the responsibility of stakeholders, which achieved high awareness among parents and caregivers [44]. Staff and volunteers in the sport and recreation sector had the capacity to mobilise and encourage parents/caregivers to engage with the Active Kids program. Some providers who serviced socially disadvantaged groups championed the program in their community to increase awareness of the Active Kids program, irrespective of the administrative ease or government support for these activities. Physical activity champions are widely recognised as “key components” of effective physical activity interventions and are often difficult to replicate when interventions are delivered at scale [3]. Research has shown that face-to-face promotion of interventions can increase the uptake of similar interventions [45]. Stakeholders that reported going above and beyond their organisational role to promote the Active Kids program were driven by the program’s goal to remove barriers and promote the program to community members who needed financial support. Future government interventions should consider allocating funding for organisations or human resources from the government to actively promote the program among disadvantaged or inactive communities and study the effect of this approach on voucher use. Other interventions such as targeted mass media campaigns may also be successful to promote financial incentive interventions, however these have not been documented [46]. Future programs should include and document the strategies employed to increase engagement of stakeholders and participants in financial incentive interventions.

### 4.4. Working Together towards a Shared Goal

There is great potential for interventions delivered by the sport and recreation sector to increase physical activity levels and improve public health [2,3,33]. Traditionally structured physical activity providers, particularly sports organisations, have focused on competition and elite performance [3,47]. The concept of health promoting sports clubs has been discussed over the past few decades but, has been underutilised in practice [2,3]. There is increasing recognition of the potential that sport and recreation organisations have to achieve health, wellbeing, inclusion, and sustainability agendas in partnership with governments [1]. Whitelaw et al. (2001) described five different models of health promotion through sports clubs, which progressively incorporate policies and practices into the daily practice of sports organisations [2,48]. The first and most passive model involves the promotion of total physical activity, in addition to structured participation. The Government-led Active Kids program appears to have improved sector cohesion and alignment toward physical activity promotion rather than the traditional performance focus. Further actions are required in NSW to build the capacity of structured physical activity providers to promote health as part of their core business [48].

### 4.5. Strengths and Limitations

This study is part of a complex pragmatic evaluation of the impact of the Active Kids program [36]. Structured physical activity providers were identified in the evaluation’s logic model as key stakeholders that would influence the effectiveness of the Active Kids program. Therefore, the Active Kids program evaluation protocol included this study focusing on the capacity of structured physical activity organisations [23]. This research is novel compared to previous research which focused primarily on participant outcomes and under-reported stakeholders’ involvement in implementation [26,27,30,31]. The multidisciplinary research team involved in the qualitative analysis had a range of experience in the program and qualitative analysis; using the Framework analysis method the researchers were able to ensure findings were trustworthy. Engagement with policy makers involved in program implementation after the interpretation of the results further strengthened the credibility of the findings against their anecdotal experiences in daily practice. This study has been critical to providing insights and learning to inform policy and practice in NSW and will provide useful guidance for similar interventions.

This study took place during the second year of the four-year Active Kids program (2018–2023) and is not without its limitations. There were over 10,000 stakeholders involved in implementing the Active Kids program across NSW; the NSW Government Office of Sport limited the recruitment process to a sample of 81 participants. The research team asked that organisations be selected from the larger database using a quota sampling technique; however, they were not involved in the selection process due to privacy constraints. The Office of Sport staff selected the 81 participants using the appropriate sampling technique; however, with only 35% of invited Active Kids providers giving consent, there is a potential for response bias. Saturation was not achieved in this study, and we acknowledge that the representative sample includes a small proportion of the total number of providers registered in the Active Kids program. Therefore, the results should be interpreted with caution and may not reflect the experiences of all Active Kids providers. Future research should monitor stakeholder experiences throughout the implementation of financial incentive interventions to understand program adaptations, the impacts of these on organisational capacity, and program maintenance in real-world contexts.

## 5. Conclusions

This qualitative study is the first to explore the stakeholder experience and organisational impact of implementing a universal financial incentive (voucher) program focused on increasing children and adolescents’ participation in structured physical activity programs. Organisations across the sport and recreation sector engaged as partners in implementation were able to adapt to the government’s requirements for administering the program; however, support for the infrastructure and finance dimensions of capacity would have been beneficial. Once administrative ease was achieved, implementation became institutionalised among Active Kids providers. Stakeholders reported that government action to increase children and adolescents’ participation in structured physical activities outside of school hours through the voucher program was acceptable, aligned with their goals, and supported its continuation. Similar government-led interventions should embed capacity-building strategies that address the dimensions of organisational capacity most substantially impacted by the new program and monitor the response of structured physical activity providers. The further identification of mutually beneficial interventions to enable more children and adolescents to participate in structured physical activity should be undertaken and implemented considering the stakeholder capacity.

## Figures and Tables

**Figure 1 ijerph-20-04081-f001:**
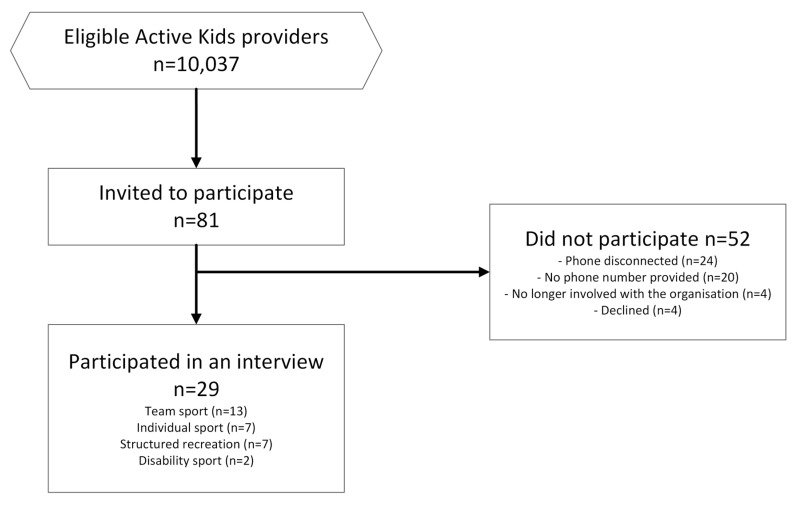
Participant recruitment flow of Active Kids providers.

**Figure 2 ijerph-20-04081-f002:**
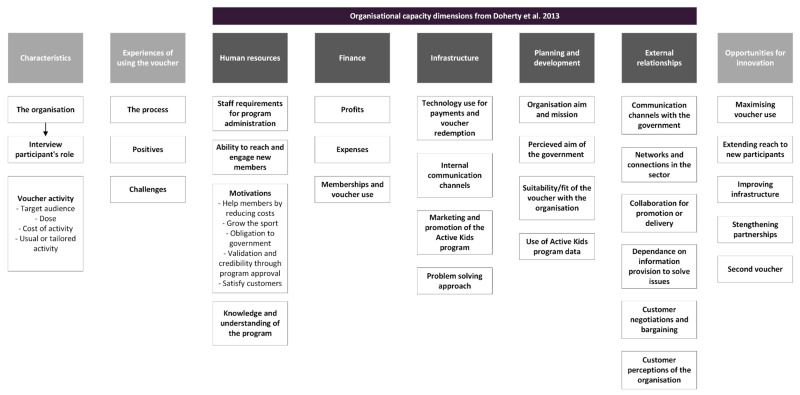
Coding framework for the interviews with Active Kids providers [19].

**Figure 3 ijerph-20-04081-f003:**
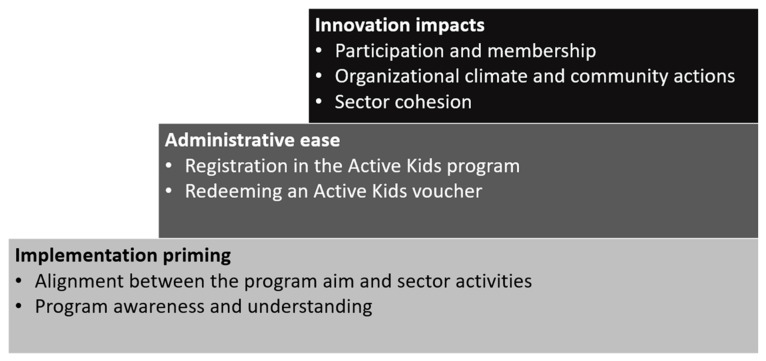
Themes identified through qualitative analysis of interviews with Active Kids providers.

**Table 1 ijerph-20-04081-t001:** Interview participant details.

Sport Type	Participant’s Role	Affiliation	Organisation Size	Location	Interview Duration
Disability sport	Committee member	Yes	Small	Regional	0:29:54
Individual sport	Committee member	Yes	Small	Regional	0:29:25
Individual sport	Committee member	Yes	Small	Regional	0:39:46
Individual sport	Committee member	Yes	Small	Regional	0:18:38
Individual sport	Committee member	Yes	Small	Metro	0:34:14
Team sport	Committee member	Yes	Small	Metro	0:35:08
Structured recreation	Committee member	Yes	Small	Regional	0:25:08
Individual sport	Registrar	Yes	Medium	Regional	0:26:51
Individual sport	Committee member	Yes	Medium	Regional	0:30:50
Team sport	Committee member	Yes	Medium	Metro	0:26:10
Structured recreation	Business owner	Yes	Medium	Metro	0:40:18
Team sport	Committee member	Yes	Medium	Metro	0:31:15
Structured recreation	Business owner	No	Medium	Metro	0:26:05
Team sport	Committee member	Yes	Medium	Regional	0:22:47
Team sport	Committee member	SSO	Large	State	0:35:09
Disability sport	Financial Director	SSO	Large	State	0:22:38
Team sport	Business owner	Yes	Large	Regional	0:39:06
Team sport	Committee member	Yes	Large	Metro	0:32:57
Team sport	Business owner	Yes	Large	Metro	0:30:01
Team sport	Program staff	Yes	Large	Regional	0:29:01
Team sport	Committee member	Yes	Large	Metro	0:26:01
Structured recreation	Program staff	No	Large	Metro	0:48:08
Structured recreation	Manager	No	Large	Regional	0:43:51
Structured recreation	Manager	No	Large	Metro	0:27:56
Team sport	Business owner	Yes	Large	Regional	0:26:53
Individual sport	Committee member	Yes	Large	Metro	0:14:35
Structured recreation	Business owner	No	Large	Metro	0:27:01
Team sport	Manager	SSO	Large	State	0:36:34
Team sport	Manager	SSO	Large	State	0:40:13

Note: Organisation size was classified based on the number of vouchers they had redeemed (small < 50; Medium 50–100, and Large > 100 vouchers).

**Table 2 ijerph-20-04081-t002:** Active Kids provider awareness of additional resources to aid implementation.

Active Kids Program Resources and Program Developments in June/July 2019	Participants Aware (total n = 29)
Adaptable Active Kids provider promotional materials, e.g., posters and graphics for use across provider’s marketing and communication platforms.	12
Online database for parents/caregivers to search for local Active Kids providers.	2
Live data dashboard showing Active Kids voucher uptake and voucher use by location.	1
Announcement of second Active Kids voucher valid July–December from July 2019.	27

## Data Availability

The dataset used and analysed during the current study is available from the corresponding author upon reasonable request.

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
