# Peer review of "“It Goes Hand in Hand with Us Trying to Get More Kids to Play” Stakeholder Experiences in a Sport and Active Recreation Voucher Program"

_ijerph, 2023, doi:10.3390/ijerph20054081_

Round 1

Reviewer 1 Report

Dear Editors and authors,

Thank you for inviting me to review this manuscript, and congratulations for your work. This is a relevant cross-sectional qualitative study that helps to understand how an Australian financial incentive (Active Kids voucher) program has impacted physical activity providers.

The present manuscript does a great job showing the different realities around this program, including small, middle size and big organisations, and metropolitan and regional ones.

Distributing the Introduction, Results and Discussion sections in different sub-sections improves the readability of the manuscript, since due to its study type, it has larger extension than average. Congratulations on this point.

Points of potential improvement:

In the keywords replacing the term “Voucher” for another one that is also suitable could be advisable, since the word “Voucher” already appears in the title, a different term would expand the indexation of the present manuscript.

In the Introduction, adding a section where physical activity and sports habits and preferences among Australian children and adolescents are presented may give a deeper context to the present work. For instances data about how many days a week do they perform physical activity and during how much time, may have interest.

Author Response

  • In the keywords replacing the term “Voucher” for another one that is also suitable could be advisable, since the word “Voucher” already appears in the title, a different term would expand the indexation of the present manuscript.
    • We appreciate this suggestion. The term voucher has been removed from the keywords, and replaced it with “financial incentive” and “fiscal”.
  • In the Introduction, adding a section where physical activity and sports habits and preferences among Australian children and adolescents are presented may give a deeper context to the present work. For instances data about how many days a week do they perform physical activity and during how much time, may have interest.
    • We appreciate this suggestion. We have added some addition text explaining the participation context in Australia.

Page 2, Lines 61-64 “Organisations in the sports and recreation sector facilitate the participation of over 3.5 million Australian children in structured physical activity outside of school time each week[13]. Australian children participate in 4-6 hours of structured physical activity each week[14].

Reviewer 2 Report

Congratulations to the authors for their work and the synthesis of the study.

The introduction is well laid out on the basis of the objective and synthesising the key ideas. The methodology clarifies well the programmes and their characteristics as well as the sample and the procedure.

The results are well presented and the discussion considers the ideas raised in the introduction.

The conclusions are coherent and novel.

Author Response

Thank you for your review and positive feedback.

Reviewer 3 Report

This seems more like quality assurance/improvement rather than research.  It is the type of report I would expect to see in gray literature rather than a journal.   I’m not sure that it contributes to research on use of taxpayer vouchers to support physical activity in children.  I found the report interesting and useful for improving the NSW voucher system, but I couldn’t see the scholarly research contribution.

Some suggestions.

The title highlights play and it is from a stakeholder quote, but it is misleading to have this in the title as the voucher system promotes sport, not play.

The numbers in Figure 1 don’t seem to be correct.  82 invited, 51 did not participate = 31, not 29.  Also, of the 51 who did not participate, the categories are 24 + 20 + 4 + 4 = 52, not 51.

Did it not seem strange that 24/82 services with families claiming government vouchers had their phone disconnected (Figure 1)?  That seems to be worthy of comment as it suggests more than a quarter of the businesses were not well-established or perhaps the business closed during the COVID-19 lockdowns.  I can see that the sample was selected in 2019, but were the stakeholders contacted before COVID or during COVID lockdowns? 

20 services did not have a phone number (Figure 1).  Were there any checks for other contacts e.g. via email?  There must have been a way for parents to contact the service.  Why rely only on phone contact.

Line 132 – It would be helpful to give a USD or Euro equivalent as international readers may not be familiar with the value of AUD.

Author Response

  • The title highlights play and it is from a stakeholder quote, but it is misleading to have this in the title as the voucher system promotes sport, not play.
    • The participant quoted in the title is referring to getting more children to play sports. We understand your concern and have modified to title. It now specifies that the voucher program is for sport and recreation.
  • The numbers in Figure 1 don’t seem to be correct. 82 invited, 51 did not participate = 31, not 29.  Also, of the 51 who did not participate, the categories are 24 + 20 + 4 + 4 = 52, not 51.
    • Thank you for identifying this typographic error. We have updated the figure with correct numbers. 81 providers were invited to participate, and 52 declined – see Figure 1.
    • The text has also been updated to reflect these values.

Page 15, Lines 633-640 “There were over 10,000 stakeholders involved in implementing the Active Kids program across NSW; the NSW Government Office of Sport limited the recruitment process to a sample of 81 participants. The research team asked that organisations were selected from the larger database using a quota sampling technique, however, were not involved in the selection process due to privacy constraints. The Office of Sport staff selected the 81 participants using the appropriate sampling technique however with only 35% of invited Active Kids providers gave consent there is a potential for response bias.”

  • Did it not seem strange that 24/82 services with families claiming government vouchers had their phone disconnected (Figure 1)? That seems to be worthy of comment as it suggests more than a quarter of the businesses were not well-established or perhaps the business closed during the COVID-19 lockdowns.  I can see that the sample was selected in 2019, but were the stakeholders contacted before COVID or during COVID lockdowns?
    • Participants were first invited by email, and followed up with a phone call, if possible. We have expanded our description of the recruitment process to explain that phone numbers entered by providers were unreliable, as identified by the reviewer from Figure 1.

Page 5, Lines 184-187 “The selected Active Kids providers were invited to participate in the study by email from the Office of Sport staff which included the participant information sheet and contact details of the research team. Those who did not respond to the email, after one reminder email, were followed up with a phone call using the phone numbers they included in the registration form. More than half of the providers (n=44) phones were disconnected, or they did not include a phone number in their registration.”

  • We have included the low response rate among invited providers as a limitation in the discussion of the paper. Page 15, Lines 633-639.
  • All participants were invited before COVID-19 restrictions occurred. COVID-19 lockdowns were not in-place until 2020 in Australia, therefore did not influence the recruitment of this study.
  • 20 services did not have a phone number (Figure 1). Were there any checks for other contacts e.g. via email?  There must have been a way for parents to contact the service.  Why rely only on phone contact.
    • As addressed earlier, we did not solely rely on phone contact. The participant recruitment was done in partnership with the government, using the data collected from participants when they registered. All participants were contacted by email, and followed up via phone, if possible.
  • Line 132 – It would be helpful to give a USD or Euro equivalent as international readers may not be familiar with the value of AUD.
    • We have added an equivalent value in USD to aid international readers.

Page 3, Line 136 – “All school-enrolled children (4.5–18 years old) that resided in NSW (2.1 million) were eli-gible for one Active Kids voucher per year valued at $100AUD ($70USD).”

Round 2

Reviewer 3 Report

Thank you for the revisions to the manuscript.  The main point raised during the review was;This seems more like quality assurance/improvement rather than research.  It is the type of report I would expect to see in gray literature rather than a journal.   I’m not sure that it contributes to research on use of taxpayer vouchers to support physical activity in children.  I found the report interesting and useful for improving the NSW voucher system, but I couldn’t see the scholarly research contribution.

I couldn't see this point had been address and the manuscript still appears to be a good report for quality assurance, but not a scholarly research contribution.

Author Response

see the attachement
